# Endometriosis-Related Chronic Pelvic Pain

**DOI:** 10.3390/biomedicines11102868

**Published:** 2023-10-23

**Authors:** Soo Youn Song, Ye Won Jung, WonKyo Shin, Mia Park, Geon Woo Lee, Soohwa Jeong, Sukjeong An, Kyoungmin Kim, Young Bok Ko, Ki Hwan Lee, Byung Hun Kang, Mina Lee, Heon Jong Yoo

**Affiliations:** 1Department of Obstetrics & Gynecology, Chungnam National University School of Medicine, Chungnam National University Sejong Hospital, 20, Bodeum 7 ro, Sejong 30099, Republic of Korea; sysong@cnuh.co.kr (S.Y.S.); wonyberry@cnuh.co.kr (Y.W.J.); bluered120@cnuh.co.kr (W.S.); 2Department of Obstetrics & Gynecology, Chungnam National University School of Medicine, Chungnam National University Hospital, 33, Munhwa-ro, Jung-gu, Daejeon 2868, Republic of Korea; mia86@cnuh.co.kr (M.P.); obgy@cnuh.co.kr (G.W.L.); tnghk8156@gmail.com (S.J.); 20210139@cnuh.co.kr (S.A.); emily6548@cnuh.co.kr (K.K.); koyoung@cnuh.co.kr (Y.B.K.); oldfox@cnuh.co.kr (K.H.L.); missinglime@cnuh.co.kr (B.H.K.); minari73@cnuh.co.kr (M.L.)

**Keywords:** endometriosis, chronic pelvic pain, inflammation, peripheral sensitization, central sensitization, cross sensitization

## Abstract

Endometriosis, which is the presence of endometrial stroma and glands outside the uterus, is one of the most frequently diagnosed gynecologic diseases in reproductive women. Patients with endometriosis suffer from various pain symptoms such as dysmenorrhea, dyspareunia, and chronic pelvic pain. The pathophysiology for chronic pain in patients with endometriosis has not been fully understood. Altered inflammatory responses have been shown to contribute to pain symptoms. Increased secretion of cytokines, angiogenic factors, and nerve growth factors has been suggested to increase pain. Also, altered distribution of nerve fibers may also contribute to chronic pain. Aside from local contributing factors, sensitization of the nervous system is also important in understanding persistent pain in endometriosis. Peripheral sensitization as well as central sensitization have been identified in patients with endometriosis. These sensitizations of the nervous system can also explain increased incidence of comorbidities related to pain such as irritable bowel disease, bladder pain syndrome, and vulvodynia in patients with endometriosis. In conclusion, there are various possible mechanisms behind pain in patients with endometriosis, and understanding these mechanisms can help clinicians understand the nature of the pain symptoms and decide on treatments for endometriosis-related pain symptoms.

## 1. Introduction

Endometriosis is one of the most common gynecological disorders in reproductive-aged women. This disease is defined as the presence of endometrial stroma and glands outside the uterus [1]. It is estimated that about 10% of women of reproductive age have this disease [2].

Endometriosis is an estrogen-dependent disease and a chronic inflammatory condition characterized by the ectopic implantation of uterine endometrial cells across multiple organs [3,4]. Retrograde menstruation is the most generally accepted early mechanism of endometriosis [5]. However, even though endometrial fragments migrate into the peritoneal cavity in about 90% of women during normal menstruation, only about 10% of women develop endometriosis [6]. It is thought that many factors, including predisposing factors and propagating factors, can play a role in the development of endometriosis [7]. And the prevalence of retrograde menstruation was similar between women with or without endometriosis [8]. First-degree relatives or twins of endometriosis are more likely to develop endometriosis, and in a more severe stage [9]. Genetic variants were found to be linked with severe endometriosis and may impact the inflammation, adhesion, growth, and hormone receptor of the lesions [10]. Anatomical factors such as outflow tract obstructions can make patients more vulnerable to the disease [11]. Other than retrograde menstruation, coelomic metaplasia, which is the metaplasia of cells into endometrial cells within the peritoneum, can explain endometriosis in prepubescent girls or at extra-pelvic sites including the thoracic cavity [12,13]. Once the process initiates, enhanced inflammatory responses, alterations in immune responses, and hormonal changes such as progesterone resistance can enhance the progression of the disease [14].

Women with endometriosis can experience cyclic pain such as dysmenorrhea, nociceptive pain such as dyspareunia, and chronic pelvic pain [15,16]. More than 60% of women who are diagnosed with endometriosis complain of chronic pelvic pain, but the mechanism behind this chronic pain is not clearly defined [3]. Chronic pelvic pain does not directly correlate with the size of the lesion or the severity of the disease [17]. In some cases, the pain remains even after the surgical removal of endometrial lesions, and chronic pain recurs in patients after 12 months [18,19]. Moreover, patients with endometriosis have many comorbid chronic pain syndromes such as irritable bowel syndrome, painful bladder syndrome, vulvar vestibulodynia, and abdomino-pelvic myalgia, suggesting a complex mechanism behind endometriosis-associated pain [20].

In this review, we explore the mechanism of the development of chronic pelvic pain in patients with endometriosis based on animal studies and clinical data.

## 2. Mechanism of Chronic Pelvic Pain in Endometriosis

### 2.1. Inflammation

Endometriosis is known to be a chronic inflammatory disease. Endometriotic lesions and the peritoneal fluid of endometriosis patients contain many inflammatory cells, cytokines, and chemokines, creating an inflammatory microenvironment [2]. The most commonly accepted theory of endometriosis is retrograde menstruation, which is the implantation of endometrial cells into the pelvic cavity during normal menstruation [21]. Normal menstruation itself is an inflammatory process that increases many tissue-resident immune cells [22]. During retrograde menstruation, inflammatory cells are recruited to newly developed ectopic endometriotic lesions [23,24]. Macrophages, mast cells, and neutrophils along with other inflammatory cells are recruited, thus enhancing the production of many inflammatory factors including interleukins (IL) such as IL-1β, IL-37, and IL-6; tumor necrosis factor (TNF)-α; nerve growth factor (NGF); and pain-associated substances such as prostaglandin, substance P, and glycodelin [25,26,27]. These inflammation-associated cytokines, chemokines, and inflammatory mediators can act on inflammatory cells, increasing the recruitment of inflammatory cells [22]. This vicious cycle can further enhance the growth and infiltration of endometriotic lesions and induce a chronic inflammatory microenvironment, and thus chronic pelvic pain [22].

Among the various immune cells that are crucial in the development of chronic inflammation in endometriosis, macrophages are recruited to remove cell debris in endometriotic lesions and promote the neovascularization of endometriosis by generating proliferation, pro-angiogenic signaling [28]. Macrophages are divided into two types: M1 macrophages, which are activated and produce pro-inflammatory cytokines and chemokines, and M2 macrophages, which are involved in anti-inflammatory responses and tissue repair. Macrophages within endometriotic lesions are M2-dominant and are involved in the growth of ectopic endometrium and tissue remodeling [29]. Macrophages can activate pro-inflammatory transcription factor NF-KB and increase the expression of TNF-α, IL-6, IL-1β, and transforming growth factor (TGF)-β at the protein level [30,31] (Figure 1). The number of mast cells and activated mast cells is increased in endometriosis, especially in deep-infiltrating endometriosis [32,33,34]. Other than releasing mediators of allergic reactions, mast cells can produce growth factors, pro-inflammatory mediators such as IL-2, IL-3, IL-6, IL-7, IL-9, IL-10, interferon-γ, TNF-α, and chemokines (CXCL8, CCL2, and CCL5) [35]. Mast cells are involved in neuropathic pain where they can directly sensitize/activate primary nociceptive neurons via mediators such as histamine leukotriene, tryptase, TNF-α, prostaglandins, serotonin, IL-1, and IL-8, or indirectly by recruiting leukocytes that can release algesic mediators [36,37,38]. The infiltration of neutrophils into the peritoneum is significantly increased in women with endometriosis [39]. Neutrophils can contribute to the growth of endometriotic lesions by producing pro-angiogenic factors and vascular endothelial growth factors [40].

IL-1β is a potent pro-inflammatory cytokine which is mainly produced by monocytes and macrophages [22]. The concentration and activity of IL-1β were reported to be increased in the peritoneal fluid, serum, and endometriotic lesions of women with endometriosis [41]. IL-1β can aggravate endometriosis-associated pain and inflammation by increasing the secretion of brain-derived neurotrophic factor in eutopic endometrial stromal cells at the mRNA and protein level [31,41]. IL-6 and IL-8 are also increased in the peritoneal fluid of endometriosis patients [42,43]. IL-6 can activate macrophages and is involved in the cellular proliferation of endometriosis, IL-8 is an angiogenic, pro-inflammatory growth-promoting cytokine involved in inflammatory responses including the activation of many inflammatory cells [44,45,46]. TGF-β1, which is increased in peritoneal fluid of patients with deep-infiltrating endometriosis, is involved in inflammatory pain and hyperalgesia [47]. Clinically, the severity of dysmenorrhea is positively correlated with increase in TGF-β1 [48]. Increased concentrations of TNF-α in endometriosis patients are involved in angiogenesis and inflammation, and the level of TNF-α along with glycodelin is associated with the severity of menstrual pain [49,50].

Prostaglandins are crucial mediators of chronic inflammation and directly generate pain [26]. They can activate nerve endings to sense pain and release other algesic mediators such as histamine, serotonin, NGF, and prostanoids from other cells or afferent nerves [51]. In endometriosis patients, increased levels of prostaglandin E2 and prostaglandin F2 α are known to be associated with both severe dysmenorrhea and dyspareunia, and noncyclic pelvic pain [27,52,53,54].

The expression of NGF is increased in endometriotic lesions, and it can increase the sprouting of nociceptors, increase the number of sensory neurons, and function in persistent inflammatory pain [55]. NGF can also increase the expression of substance P and calcitonin gene-related peptide, thus modulating the central transmission of pain [51]. Monocyte chemotactic protein-1, which exists in high levels in the peritoneal fluid of endometriosis patients, can facilitate increases in activated macrophages in endometriosis and is involved in endometrial cell proliferation and the secretion of other cytokines and growth factors via mast cell activation [56,57].

Reactive oxygen species, which are a byproduct of oxidative stress, are also increased in endometriosis [58,59]. Oxidative stress can alter the nociceptive transient receptor potential vanilloid 1 (TRPV1) cation channel, subfamily V, member 1, which is associated with pain generation in inflammatory conditions [60,61,62].

### 2.2. Innervation

The aberrant innervation of endometriotic lesions is considered pivotal in the role of chronic pelvic pain in endometriosis patients. Pain is a process in which noxious stimuli are recognized at the level of peripheral nerve fibers called nociceptors and transmitted to the spinal cord and brain [63]. Ectopic endometriotic lesions disseminated into the peritoneal cavity do not have an intrinsic nerve supply, meaning that new nerve fibers have to form in order to transmit pain from the endometriotic lesion [64]. The density of nerve fibers, including Aγ sensory, C sensory, cholinergic, and adrenergic nerve fibers, is greater in peritoneal endometrial tissue compared to healthy peritoneum [47,65,66]. One study found that peritoneal fluid from endometriosis patients increased the sprouting of sensory neurites from the dorsal root ganglia [65]. The author hypothesized that NGF and IL-1β in the peritoneal fluid could cause an overbalance of substance P-positive nerve fibers, thus creating a pro-inflammatory milieu that can affect peritoneal fluid in turn [65].

Although some studies reported increased nerve fibers in ovarian endometrioma compared to healthy ovaries, they did not evaluate pain [67]. Another study did not find increased nerve fibers in ovarian endometrioma [68]. This may have been due to the lower correlation of ovarian endometrioma with pain compared to peritoneal endometriosis or deep-infiltrating endometriosis.

Women with endometriosis with higher pain scores for dysmenorrhea and pelvic pain had significantly increased concentrations of neuronal markers (neurofilament and protein gene product), and severe dysmenorrhea was positively correlated with endometriosis-associated nerve fibers [66,68]. Increased NGFs in endometriosis patients were associated with an increased level of dense nerve supply, which was associated with severe pain [69].

Deep-infiltrating endometriosis occurs commonly in anatomical sites with rich innervations such as the rectovaginal septum, pararectal space, uterosacral ligament, rectum, and ureter, and it is more likely associated with pain than peritoneal endometriosis or ovarian endometrioma [70,71,72,73]. Rectal endometriotic lesions showed six-times-higher nerve fiber density compared to normal controls, and increased levels of the neurite outgrowth marker GAP-43 [22,25]. In enteric endometriosis, macrophage colony stimulating factor-1 produced by enteric neurons was reported to alter the inflammatory responses of muscularis macrophages near endometriotic lesions [74]. The early recruitment of macrophages in endometriosis can increase the secretion of macrophage colony stimulating factor-1, which stimulates macrophage survival and subsequent inflammation [74]. As inflammation aggravates, changes in gut microbiota and the inflammatory microenvironment can further increase the concentration of prostaglandins, which in turn exacerbate pain symptoms [74]. Moreover, enteroendocrine cells such as neuropod cells can send signals from the gut to the brainstem in a millisecond, and this rapid transmission can be the source of severe pain in deep-infiltrating endometriosis patients [75].

Aside from changes in sensory nerve fibers, the autonomic nervous system is also considered to be related to endometriosis-associated pain [22]. The autonomic nervous system which includes sympathetic and parasympathetic nervous systems is a crucial mechanism that maintains the homeostasis of the organism [22]. The autonomic nervous system not only modulates vascular/nonvascular smooth-muscle contractile activity, intestine movement, glandular secretion, and immune cell interaction but also transmits information about the internal environment and potential noxious stimuli to the central nervous system [75,76]. Many studies speculated that an imbalance between sympathetic, parasympathetic, and sensory innervation, as well as the abnormal secretion of cytokines, can mediate neurogenesis in endometriosis and induce peripheral neuroinflammation [47]. The density of sensory nerve fibers was increased in peritoneal endometriosis, whereas sympathetic nerve density was decreased [65,73,77]. One study showed that the peritoneal fluid from endometriosis patients decreased the neurite outgrowth from sympathetic ganglia [65]. Although the exact mechanism has not been determined, alterations in the structure and function of sympathetic nerve fibers are associated with chronic inflammation as evident in other chronic inflammatory diseases such as inflammatory bowel syndrome and rheumatoid arthritis [47,73,78,79]. Aside from inflammation, alterations in endometriotic innervation might cause changes in a whole pelvic cavity innervation itself [22]. These changes, along with alterations of the neural network, might affect the progression of peripheral neuropathic pain [80].

In addition to numbers and the distribution of nerve fibers, the level of neurotrophins is also closely related to endometriosis-associated pain. Alterations in NGFs, which are crucial mediators of pain and inflammation, are positively correlated with nerve fiber density and neuropathic pain [65]. NGF/TrkA signaling, involved in neurodegenerative diseases such as Alzheimer’s disease, chronic pain, inflammation, and cancer, is closely related to neuronal development, growth, survival, and function [81]. NGF increases the expression of neuropeptides that can modulate central pain transmission, substance P, and calcitonin gene-related peptides [51]. NGF can increase the sensory neurons that are involved in pain sensation and is related to hyperalgesia or persistent inflammatory pain [55]. An increased expression of NGF and TrkA was found in patients with endometriosis [82], and immunointensity of NGF in the stroma is highly associated with local nerve fiber density and the severity of deep dyspareunia [25,83]. BDNF is another neurotrophin that is increased in patients with endometriosis [84,85]. BDNF is involved in neuronal cell growth, survival, and differentiation [86]. BDNF is also related to inflammation and was shown to play a pro-nociceptive role in an inflammatory model, enhancing the hyperalgesic response [87].

Nonetheless, growth factors other than neurotrophins such as vascular endothelial growth factor and TGF-β are also increased in the peritoneal fluid of women with endometriosis [20]. An increase in these growth factor can give rise to the growth of newly developed nerve cells and contributes to the increase in nerve fiber density in endometriotic lesions [82].

### 2.3. Endometriosis and Peripheral Sensitization

Women with endometriosis often experience chronic pelvic pain that persists even after the removal of the endometriotic lesion [17]. Peripheral sensitization is one mechanism that can explain the persistence of pain even without lesions [20] (Figure 2). Nociceptors can become more susceptible to pain after local tissue injury or the inflammatory process, due to neuroplasticity of the peripheral sensory nerve. The process of pain sensitization includes a decreased threshold to firing at peripheral sensory neurons, an increased response to stimuli, and expansion of receptive fields [88,89]. This peripheral sensitization can be influenced by many factors including changes in sensory nerve density, the autonomic nervous system, neuroinflammation, changes in many receptors and ion channels, and perineural invasion [20].

Peripheral sensitization is most commonly thought to be caused by inflammatory changes in the chemical environment of nerve fibers [90]. As mentioned above, inflammatory cells such as mast cells, macrophages, neutrophils, neurotrophins, cytokines, and chemokines are all increased in endometriotic lesions [25,31,91,92,93,94]. Nociceptors have more than one cell surface receptor that can recognize and respond to pro-inflammatory and pro-algesic agents [89]. These interactions can increase the excitability and sensitivity of nerve fibers [89]. NGF not only promotes neuronal survival and development but also acts on the peripheral nociceptor terminal, most notably TRPV1 to increase sensitivity via NGF/TrkA interaction [95]. NGF can transport back to the nociceptor nucleus to increase the expression of nociceptive protein substance P, TRPV1, and Nav1 voltage-gated Na channel subunit [96,97]. Like other inflammatory pain diseases in which TRPV1 is correlated with pain [98,99], the expression of TRPV1 was increased at the dorsal root ganglion rats with endometriosis [100]. Increased levels of TRPV1 in patients with endometriosis were positively associated with pain intensity [101,102]. Increased levels of IL-1β, IL-6, or TNF-α were shown to augment the production of prostaglandins, NGF, and bradykinin, thus promoting hypersensitivity [89]. Increased levels of TNF-α and glycodelin were associated with hyperexcitability to electrical stimulation and severe menstrual pain in women with endometriosis [49,103]. And TNF-α levels were associated with severe pain regardless of the size of the lesion [104].

Perineural invasion, as described in gynecologic cancer, is the spread of the lesion through the lumbosacral plexus and pelvic autonomic nerve to peripheral nerves [105]. Many studies have reported perineural invasion in patients with endometriosis [82,106]. Of them, 95% with peripheral nerve spread had pain symptoms [107], and perineural involvement in patients with deep-infiltrating endometriosis was positively correlated with the severity of chronic pelvic pain [71]. Perineural invasion is also associated with neurogenesis and angiogenesis [64], and NGF and TGF-α can promote lesional perineural invasion [48,69,82].

### 2.4. Endometriosis and Central Sensitization

Women with endometriosis experience from severe pain that does not correlate with the severity of the disease and often experience hyperalgesia in parts of the body outside the pelvis [17,108,109,110,111]. Central sensitization seems to play an important role in converting nociception to chronic pain [112,113]. Central sensitization is defined as the “increased responsiveness of nociceptive neurons in the central nervous system to their normal or subthreshold afferent input” [114]. When exposed to continuous peripheral stimuli, neuronal circuits within the nociceptive pathway can change, increasing their responses to noxious stimuli or exhibiting a pain response to innocuous stimuli [115,116,117,118]. The altered synaptic efficacy of neighboring nerves can promote central facilitation that can recognize innocuous stimuli as pain [20]. Dysfunctional descending pain modulation at the level of the dorsal horn neuron can also contribute to central sensitization [119].

In an animal model of endometriosis, increased allodynia or hyperalgesia was reported to noxious stimuli such as heat or vaginal distention that are unrelated to endometriosis [120,121,122]. Similarly, women with endometriosis showed enhanced muscular pain after a saline injection to their hands compared to healthy controls [115]. Women with endometriosis-associated chronic pain also showed increased hyperalgesia and allodynia [123].

Central sensitization can also occur due to changes in brain activity or structure. Alterations of brain activity in women with chronic pain have been studied using functional magnetic resonance imaging and positron emission tomography imaging. Women with dysmenorrhea showed increased sensitivity to noxious thermal stimuli compared to women without dysmenorrhea [124,125]. Communication between brain regions was altered in endometriosis. Women with endometriosis pain showed increased resting connectivity between the anterior insula, which is the major pain processing region, and other regions of the brain compared to healthy controls or endometriosis patients without pain [126]. Women with painful endometriosis also showed higher excitatory neurotransmitters in the anterior insula compared to the other two groups [126]. Increases in neurotransmitters were associated with connectivity between the anterior insula and the medial prefrontal cortex, which is a pain modulatory region [126]. Dysfunction in endogenous pain inhibition, which is modulated by many spinal and cortical mechanisms, could also enhance the development of chronic pain [47,127]. In adolescents with endometriosis, functional connectivity between the anterior insula and the cerebellum was positively correlated with the self-reported pain intensity level [128]. While functional magnetic resonance imaging and positron emission tomography imaging can indirectly investigate brain function by measuring metabolic activity, electrical activity of the brain in women with endometriosis has been also investigated using an electroencephalogram [47]. In women with endometriosis-related chronic pelvic pain, pain-related network connectivity was altered, and endometriosis patients showed increased somatosensory cortex connectivity, which indicates sustained activation of the somatosensory pain system [129]. Endometriosis-related chronic pelvic pain was associated with increased amplitudes of resting delta and beta waves, which could be associated with cholinergic tone and stress reactivity [130].

Alterations in the structure of specific brain regions were also studied in patients with chronic pain [131]. Women with chronic pelvic pain showed decreased grey matter volume in the areas involved in pain perception such as the thalamus, cingulate gyrus, putamen, and insula [132]. It is hypothesized that neuronal atrophy induced by the neurotoxic effect of repeated pain, alterations in neurotransmitter concentrations or the metabolic activity of the neuron, neurodegeneration caused by pain-related inactivity, psychologic factors, comorbidities, and medications can induce those volume changes [47]. Not only neuronal changes but also glial adaptation have roles in central adaptation. Spinal glial adaptation as well as increases in microglial soma size in the brain cortex hippocampus, thalamus, and hypothalamus were observed in a mouse model of endometriosis [133,134].

Although not direct evidence, changes in the hypothalamus–pituitary–adrenal (HPA) axis can be associated with central sensitization. Alterations in the HPA axis and HPA-axis-mediated pain response are seen in patients with persistent pelvic pain. Women with dysmenorrhea showed decreased cortisol levels compared to women without pain [135,136]. This result might be because acute stress activates the HPA axis, but chronic insults attenuate this response [20].

A central sensitization assessment tool estimated that patients with endometriosis showed more than a 40% prevalence of central sensitization [137]. High central sensitization scores were associated with deep-infiltrating endometriosis and poor postoperative pain outcomes [138,139].

### 2.5. Cross Sensitization

Women with endometriosis have high comorbidities with other chronic pain syndromes related to nervous system sensitization such as irritable bowel syndrome, painful bladder syndrome, and vulvar vestibulodynia [140]. Women with endometriosis have about 43–60% of coexisting bladder pain syndrome, which is a pain in the bladder that is associated with urinary symptoms such as urgency and frequency [141,142]. The rate of irritable bowel syndrome, which is a functional gastrointestinal disorder with an abdominal pain related to a change in bowel habit, is as high as 60% in women with endometriosis [143,144], and vulvodynia coexists with endometriosis in 11% of women [145,146]. Cross sensitization occurs as nociceptive input from diseased tissue affects the pain perception of neighboring normal tissue [147]. The exact mechanism of cross sensitization has not been identified, but the overlap of the peripheral afferent pathway within the dorsal root ganglia and the spinal cord is thought to be a crucial part of the mechanism [148,149]. Visceral afferents from not only the uterus but also the bladder and colon converge at a similar site of the spinal cord, and therefore sensitize the adjacent cells due to the spatial location [150,151]. Dichotomizing afferents, which are single peripheral neuronal cell bodies that can generate multiple axons to innervate different abdominal organs simultaneously, are also suggested as a mechanism for cross sensitization [152,153,154]. These multiple nerve fibers from multiple organs gather into a single cell body within the dorsal root ganglion, thus affecting each other. These shared pathways can coordinate pelvic organ functions such as urination, defecation, and sexual functions. However, in endometriosis, these pain pathways can be sensitized in a pathologic condition that allows cross organ sensitization to occur [3]. Viscero-visceral hyperalgesia, which is an increased pain in women with endometriosis and the associated pain syndrome, is thought to be associated with endometriosis pain severity [154].

### 2.6. Psychosocial Factors

Pain intensity can be affected by psychologic conditions such as depression, anxiety, pain catastrophizing, pain expectation, and attention to pain [155,156,157,158,159]. Depression can disrupt normal emotion regulatory circuits [155]. Other than endometriosis, depression is highly associated with pain score in fibromyalgia and rheumatoid arthritis in certain brain areas [160,161]. Anxiety seems to amplify pain through a hippocampal network. Catastrophizing pain might affect descending inhibitory pathways [156]. Almost 90% of women with endometriosis experience psychological problems such as depression and anxiety [162,163]. Moreover, women with endometriosis showed significantly higher pain catastrophizing scores [164]. Even though the direct mechanism between psychological factors and endometriosis-associated chronic pain has not been identified, their close relationship needs further research as psychological factors not only enhance pain in these women but also decrease the quality of life.

## 3. Conclusions

Chronic pelvic pain is a life-debilitating disease that can significantly decrease the quality of life for women with endometriosis. The pathophysiology behind endometriosis-associated chronic pelvic pain is very complex and remains to be clarified. Data from animal studies and the samples from endometriosis patients suggest that altered inflammation including inflammatory microenvironment, increased vascular and neuronal growth factors, and reactive oxygen species can generate chronic pelvic pain in endometriosis. Increased density of nerve fibers, altered autonomous nervous system, and increased level of neurotrophins also are believed to play pivotal parts. Other than local factors, nervous system sensitization is a crucial part of chronic pelvic pain in endometriosis. Peripheral sensitization augmented by neuroinflammation and perineural invasion, central sensitization induced by alterations in the brain activity and the H-P-A axis, and cross sensitization all contribute to chronic pelvic pain in endometriosis. Understanding the mechanism behind this complex condition can aid physicians in better understanding patients and ultimately develop an effective treatment modality to alleviate symptoms of patients who suffer from chronic pelvic pain associated with endometriosis.

## Figures and Tables

**Figure 1 biomedicines-11-02868-f001:**
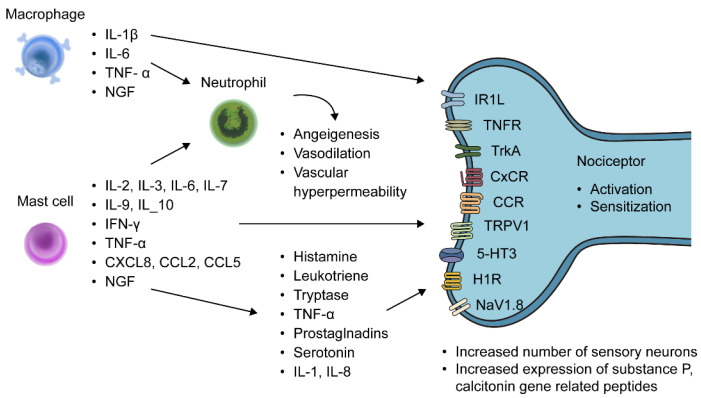
Involvement of macrophages and mast cells in the generation of pain associated with endometriosis. Macrophages can release inflammatory mediators including TNF-α, IL-6, IL-1β, and TGF–β, and nerve growth factors. Mast cells can release inflammatory cytokines including IL-2, IL-3, IL-6, IL-7, IL-9, IL-10, IFN-γ, TNF-α, and chemokines (CXCL8, CCL2, and CCL5) and nerve growth factor. These inflammatory mediators recruit neutrophils that induce angiogenesis, vasodilation, and vascular hyperpermeability. Mast cells can sensitize nociceptive neurons via releasing mediators such as leukotriene, histamine, tryptase, TNF-α, prostaglandins, and serotonin. Mediators released from macrophages, mast cells, and other inflammatory cells can activate their receptors ILR, TNFR, TrkA, CXC chemokine receptor, and C-C motif chemokine receptors expressed on the nociceptive neurons. NGF from mast cells and macrophages can increase the number of sensory neurons and expression of pain-related mediators such as substance P and calcitonin gene-related peptide. IL, interleukin; TNF, tumor necrosis factor; NGF, nerve growth factor; IFN, interferon; CXCL, C-C motif chemokine ligand; CCL, C-C motif chemokine ligand; IL1R, interleukin 1 receptor; TNFR, tumor necrosis factor receptor; TrkA, tropomyosin receptor kinase A; CXCR, CXC chemokine receptor; CCR, C-C motif chemokine receptor; TRPV, transient receptor potential vanilloid 1 cation channel subfamily V member; 5-HT3, 5-hydroxytryptamine receptor; H1R, istamine-1 receptor; NaV1.8, voltage-gated sodium ion channel subtype.

**Figure 2 biomedicines-11-02868-f002:**
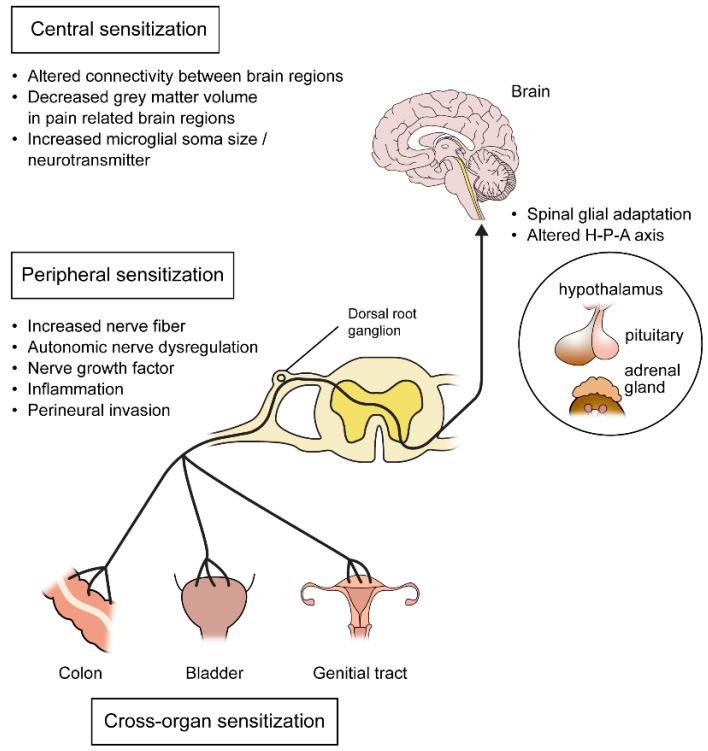
Peripheral, central, and cross-organ sensitization of the nervous system in patients with endometriosis. H-P-A, hypothalamus–pituitary–adrenal.

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
