# Peer review of "Endometriosis-Related Chronic Pelvic Pain"

_biomedicines, 2023, doi:10.3390/biomedicines11102868_

Round 1

Reviewer 1 Report

Dear Authors,

In the paper you reported the fisiopathological mechanism of the chronic pelvic pain related to endometriosis.

Unfortunately, your paper doesn’t add nothing of new to the current literature.

Anyway, you should change the type of article from review to communication because the paper has not criteria neither for systematic review (no statistical data) neither for narrative review.

In my opinion the quality of the paper it’s not enough for publication.

nothing 

Author Response

  • We appreciate your thoughtful comment. We agree with your valuable view that there are already lots of published data regarding this issue. We tried to focus on chronic pain associated with endometriosis and tried to add some more recent data. According to your comment, we tried to revise the manuscript with adding some more figures to demonstrate more clearly the aspect of this paper. Again, thank you for your valuable time reviewing this paper.
  • <In Page 2>
  • Figure 1. Schematic figure of possible mechanisms involved in generation of chronic pain associated with endometriosis.
  • TNF, tumor necrosis factor; IL, Interleukin; NGF, nerve growth factor; TGF, transforming growth factor; CXCL1, C-C motif chemokine ligand 1; CCL, C-C motif chemokine ligand; PG, prostaglandin; VEGF, vascular endothelial growth factor; TRPV, transient receptor potential vanilloid 1 cation channel subfamily V member; TrkA, tropomysin receptor kinase A; NaV1.8, voltage gated sodium ion channel subtype; H-P-A, hypothalamus-pituitary-adrenal

<In Page 3>

Figure 2. Involvement of macrophage and mast cells in generation of pain associated with endometriosis. Macrophages can release inflammatory mediators including TNF- α, IL-6, IL-1β, and TGF–β, and nerve growth factors. Mast cells can release inflammatory cytokines including IL-2, IL-3, IL-6, IL-7, IL-9, IL-10, IFN-γ, TNF- α, and chemokines (CXCL8, CCL2, and CCL5) and nerve growth factor. These inflammatory mediators recruit neutrophils that induce angiogenesis, vasodilation and vascular hyperpermeability. Mast cells can sensitize nociceptive neurons via releasing mediators such as leukotriene, histamine, tryptase, TNF- α, prostaglandins, serotonin. Mediators released from macrophages, mast cells and other inflammatory cells can activate their receptors ILR, TNFR, TrkA, CXC chemokine receptor, C-C motif chemokine receptors expressed on the nociceptive neurons. NGF from mast cells and macrophages can increase the number of sensory neurons and expression of pain related mediators such as substance P and calcitonin gene related peptide.

IL, Interleukin; TNF, tumor necrosis factor; NGF, nerve growth factor; IFN, interferon; CXCL, C-C motif chemokine ligand; CCL, C-C motif chemokine ligand; VEGF, vascular endothelial growth factor; H1R, istamine-1 receptor;TRPV, transient receptor potential vanilloid 1 cation channel subfamily V member; 5-HT3, 5-hydroxytryptamine receptor ;TrkA, tropomysin receptor kinase A; TNFR, tumor necrosis factor receptor ;NaV1.8, voltage gated sodium ion channel subtype

<In Page 6>

Figure 3. Peripheral, central, and cross-organ sensitization of nervous system in patients with endometriosis.

H-P-A, hypothalamus-pituitary-adrenal

Reviewer 2 Report

By analyzing both animal studies and clinical data, in the present review the authors explored the mechanism leading to one of the most common aspects of endometriosis: the chronic pelvic pain. This aspect is very interesting and the review

However, the manuscript, while developing a topical and interesting theme, is too flat and enunciative. It could be much improved by schematizing and focusing on the main points of interest.

Chapter 2

·       At the beginning of the chapter a figure of a brief concise scheme summing up the involvement of the potential cells, with the relative inflammatory cytokines, involved in the process should be added, thus introducing all the following sub-chapters for a better focusing of what induces what.

·       A second figure that describes in more detail the activity of macrophages and mast cells in inducing the process, not only of inflammation, which is now well known, but also that of pain activation, would bring great news to the focus of the review

·       Same for the interesting sub-chapters 2.2-2.5 : the authors should add clear pictures describing the mechanism of the increased nerve formation, the brain inter-communications, and the cross linking leading to multiple organ sensitization, should be added. A clear description of how peripheral sensitization, by binding to the central nervous system, decreases the pain threshold by sensitizing the affected areas, even after the problem has been surgically eliminated, would result emphasized.

Author Response

  • We deeply appreciate your thoughtful comments. We totally agree with your valuable suggestions that we need figures to clearly emphasize and focus on the issue of this paper. According to your valuable comment, we have added three figures to more clearly show the focus of this paper.; first on brief scheme summing up the involvement of the potential cells, relative inflammatory cytokines and other mechanisms involved in the pain generation in endometriosis, second on the activity of macrophages and mast cells in inducing pain, third on the nervous system sensitization. Again, thank you for your valuable time reviewing this paper, and valuable suggestions.

<In Page 2>

Figure 1. Schematic figure of possible mechanisms involved in generation of chronic pain associated with endometriosis.

TNF, tumor necrosis factor; IL, Interleukin; NGF, nerve growth factor; TGF, transforming growth factor; CXCL1, C-C motif chemokine ligand 1; CCL, C-C motif chemokine ligand; PG, prostaglandin; VEGF, vascular endothelial growth factor; TRPV, transient receptor potential vanilloid 1 cation channel subfamily V member; TrkA, tropomysin receptor kinase A; NaV1.8, voltage gated sodium ion channel subtype; H-P-A, hypothalamus-pituitary-adrenal

<In Page 3>

Figure 2. Involvement of macrophage and mast cells in generation of pain associated with endometriosis. Macrophages can release inflammatory mediators including TNF- α, IL-6, IL-1β, and TGF–β, and nerve growth factors. Mast cells can release inflammatory cytokines including IL-2, IL-3, IL-6, IL-7, IL-9, IL-10, IFN-γ, TNF- α, and chemokines (CXCL8, CCL2, and CCL5) and nerve growth factor. These inflammatory mediators recruit neutrophils that induce angiogenesis, vasodilation and vascular hyperpermeability. Mast cells can sensitize nociceptive neurons via releasing mediators such as leukotriene, histamine, tryptase, TNF- α, prostaglandins, serotonin. Mediators released from macrophages, mast cells and other inflammatory cells can activate their receptors ILR, TNFR, TrkA, CXC chemokine receptor, C-C motif chemokine receptors expressed on the nociceptive neurons. NGF from mast cells and macrophages can increase the number of sensory neurons and expression of pain related mediators such as substance P and calcitonin gene related peptide.

IL, Interleukin; TNF, tumor necrosis factor; NGF, nerve growth factor; IFN, interferon; CXCL, C-C motif chemokine ligand; CCL, C-C motif chemokine ligand; VEGF, vascular endothelial growth factor; H1R, istamine-1 receptor;TRPV, transient receptor potential vanilloid 1 cation channel subfamily V member; 5-HT3, 5-hydroxytryptamine receptor ;TrkA, tropomysin receptor kinase A; TNFR, tumor necrosis factor receptor ;NaV1.8, voltage gated sodium ion channel subtype

<In Page 6>

Figure 3. Peripheral, central, and cross-organ sensitization of nervous system in patients with endometriosis.

H-P-A, hypothalamus-pituitary-adrenal

Reviewer 3 Report

The paper seems well written and constructed as a review article , and has some value for readers to understand the mechanism of chronic pelvic pain with endometriosis. This is why I think it is acceptable in the present form.

Author Response

  • We are very grateful for your kind and thoughtful remark on this manuscript. We have added three figures to improve the manuscript. Again, we deeply appreciate your precious time spent reviewing this paper.

<In Page 2>

Figure 1. Schematic figure of possible mechanisms involved in generation of chronic pain associated with endometriosis.

TNF, tumor necrosis factor; IL, Interleukin; NGF, nerve growth factor; TGF, transforming growth factor; CXCL1, C-C motif chemokine ligand 1; CCL, C-C motif chemokine ligand; PG, prostaglandin; VEGF, vascular endothelial growth factor; TRPV, transient receptor potential vanilloid 1 cation channel subfamily V member; TrkA, tropomysin receptor kinase A; NaV1.8, voltage gated sodium ion channel subtype; H-P-A, hypothalamus-pituitary-adrenal

<In Page 3>

Figure 2. Involvement of macrophage and mast cells in generation of pain associated with endometriosis. Macrophages can release inflammatory mediators including TNF- α, IL-6, IL-1β, and TGF–β, and nerve growth factors. Mast cells can release inflammatory cytokines including IL-2, IL-3, IL-6, IL-7, IL-9, IL-10, IFN-γ, TNF- α, and chemokines (CXCL8, CCL2, and CCL5) and nerve growth factor. These inflammatory mediators recruit neutrophils that induce angiogenesis, vasodilation and vascular hyperpermeability. Mast cells can sensitize nociceptive neurons via releasing mediators such as leukotriene, histamine, tryptase, TNF- α, prostaglandins, serotonin. Mediators released from macrophages, mast cells and other inflammatory cells can activate their receptors ILR, TNFR, TrkA, CXC chemokine receptor, C-C motif chemokine receptors expressed on the nociceptive neurons. NGF from mast cells and macrophages can increase the number of sensory neurons and expression of pain related mediators such as substance P and calcitonin gene related peptide.

IL, Interleukin; TNF, tumor necrosis factor; NGF, nerve growth factor; IFN, interferon; CXCL, C-C motif chemokine ligand; CCL, C-C motif chemokine ligand; VEGF, vascular endothelial growth factor; H1R, istamine-1 receptor;TRPV, transient receptor potential vanilloid 1 cation channel subfamily V member; 5-HT3, 5-hydroxytryptamine receptor ;TrkA, tropomysin receptor kinase A; TNFR, tumor necrosis factor receptor ;NaV1.8, voltage gated sodium ion channel subtype

<In Page 6>

Figure 3. Peripheral, central, and cross-organ sensitization of nervous system in patients with endometriosis.

H-P-A, hypothalamus-pituitary-adrenal

Round 2

Reviewer 1 Report

Unfortunately, as suggested in the preview revision, in my opinion the paper doesn’t reach the criteria to be published.

nothing

Reviewer 2 Report

The authors greatly improved the manuscript, which is now definitely eligible to be published